# Cluster and Principal Component Analyses of the Bioactive Compounds and Antioxidant Activity of Celery (*Apium graveolens* L.) Under Different Fertilization Schemes

**DOI:** 10.3390/foods13223652

**Published:** 2024-11-17

**Authors:** Anita Milić, Boris Adamović, Nataša Nastić, Aleksandra Tepić Horecki, Lato Pezo, Zdravko Šumić, Branimir Pavlić, Milorad Živanov, Nemanja Pavković, Đorđe Vojnović

**Affiliations:** 1Faculty of Technology Novi Sad, University of Novi Sad, Bulevar cara Lazara 1, 21000 Novi Sad, Serbia; anitavakula@uns.ac.rs (A.M.); tepical@uns.ac.rs (A.T.H.); sumic@uns.ac.rs (Z.Š.); bpavlic@uns.ac.rs (B.P.); 2Faculty of Agriculture, University of Novi Sad, Trg Dositeja Obradovića 8, 21000 Novi Sad, Serbia; nemanja.pavkovic@polj.edu.rs (N.P.); djordje.vojnovic@polj.uns.ac.rs (Đ.V.); 3Institute of General and Physical Chemistry, University of Belgrade, Studentski trg 12/V, 11158 Belgrade, Serbia; latopezo@yahoo.co.uk; 4Institute of Field and Vegetable Crops, National Institute of the Republic of Serbia, Maksima Gorkog 30, 21000 Novi Sad, Serbia; milorad.zivanov@ifvcns.nc.ac.rs

**Keywords:** celery production, bioactive compounds, antioxidant activity, PCA, cluster analysis

## Abstract

This research investigates the impact of various fertilization methods on the bioactive compound content and antioxidant activity of celery (*Apium graveolens* L.) root and leaf. Mineral fertilizer, poultry manure, cattle manure, sheep manure, supercompost, and molasses were applied. Total dry weight, phenolic and flavonoid compounds, and antioxidant activity were assessed, along with fiber, protein, fat, sugar, and starch in celery root. Principal component analysis (PCA) and cluster analysis were used to correlate production conditions with the parameters. The highest fiber and protein contents were found in mineral-fertilized roots, while total fat and sugar were highest in cattle-manure-fertilized roots, and starch was highest in supercompost-fertilized roots. Fertilization with supercompost yielded the highest total phenolic and flavonoid contents in leaves, while mineral fertilizer resulted in the highest antioxidant activity in roots. Notably, the highest dry weight in leaves and the highest total phenolic and flavonoid contents in roots were also observed with supercompost. PCA and cluster analysis demonstrated significant correlations between plant parts, i.e., the celery root and leaf samples, cultivation conditions, and the observed parameters, emphasizing the importance of selecting suitable cultivation methods to optimize celery’s nutritional properties. Also, these findings suggest that supercompost, a byproduct of breweries, could potentially replace animal-based organic fertilizers, addressing the problem of reduced availability due to declining livestock numbers.

## 1. Introduction

Celery (*Apium graveolens* L.) is a plant from the *Apiaceae* family. It is an annual/perennial plant that grows throughout Europe and temperate-climate regions of Africa and Asia [1]. Celery has also been used in traditional medication to manage stomach aches and spasms, as well as a laxative, diuretic, and sedative, with many different pharmacological activities, such as antimicrobial, antifungal, antioxidant, antidiabetic, anti-infertility activities [2]. Celery is an important source of natural active products that function differently depending on mechanisms and biological properties [3]. Among the phytochemical compounds in celery, one can mention carbohydrates, phenols such as flavonoids, alkaloids, and steroids [4]. Free radicals which possess antioxidant activity are the result of the presence of various phytochemical compounds, especially polyphenols such as flavonoids, and phenolic acids [5]. Flavonoids and other phenolic compounds have biological effects, such as antioxidant activities, and are inductors for restraining free radicals and peroxidation [3].

Vegetables are herbaceous plants that accumulate huge amounts of biomass within a short duration. Thus, they demand readily available nutrients throughout the growth period [6]. Appropriate celery nutrition is very important for high-yielding celery root and leaves of good quality. Fertilization is one of the most important factors limiting plant productivity [7] and affecting the nutritional value of vegetables [8]. The type and amount of fertilizer, as well as the level of application, directly influence nutrient availability in plants and indirectly impact plant physiology and the biosynthesis of secondary compounds [8]. Organic fertilizers have an important role in plant growth, as a source of all the necessary macro and microelements in available forms during mineralization [9]. Organic fertilizers need a longer period to convert nutrients into accessible forms. The rate of mineralization of organic fertilizers depends on the type of fertilizer, and their values as sources of nutrients for plants are very different [10]. Manures are typically applied to soil at rates adequate to supply a crop’s nitrogen (N) requirement [11]. Different types of compost are highly nutritive, increasing soil’s physical, chemical, and biological properties and improving its natural fertility, but they differ in quality [12]. Fertilization is the most important and controllable factor affecting the nutritional value of vegetables. The type and value of fertilizer and the level of application directly influence the level of nutrients available in plants and indirectly influence plant physiology and the biosynthesis of secondary compounds in plants [8]. In recent times, consumers have demanded higher-quality and safer food, and are highly interested in organic vegetables [8].

Vegetable crops grown with solely organic sources of nutrients recorded lower yields, but net returns were higher [13]. Chemical fertilizers have a faster nutritive effect and are beneficial to plant growth and yield [9]. According to [14], chemical fertilizer increases the yield while organic manure improves the quality. Moyin-Jesu [15] indicates that the problems with the continuous application of mineral fertilizers are unbalanced nutrition, increased soil acidity, the degradation of physical properties, and the loss of organic matter. Organically grown foods are perceived as being of better quality, healthier, and more nutritious than their conventional counterparts [16]. Pavla and Pokluda [17] state that different fertilizers have varying effects on the nutritional value of vegetables and that the influence of organic fertilizers on the nutritional value of vegetables can be positive.

In order to provide crop supplies with nutrients and attain high yields, farmers have almost completely relied on applying mineral fertilizers. Hence, food prices and fertilizer costs are strongly correlated [18]. Since the cost of mineral fertilizers is extremely high, and the quantity of manure that can be used for vegetable farming is decreasing because the number of livestock is decreasing every year, the question arises about the possibility of using different organic materials that are created as a residue in the food industry. Using organic soil amendments as nutrient sources for vegetable crops decreases the cost of purchased inorganic fertilizers, enhances soil characteristics, improves the quality of produce, and helps to achieve sustainability in the production system [6].

The main goal of this research was to obtain a comprehensive overview of the bioactive compound content and antioxidant activity of celery root and leaf, produced in Serbia under different fertilizing regimes. Mineral fertilizer, poultry manure, cattle manure, sheep manure, supercompost, and molasses were used as fertilizer. Total dry weight, total phenolic and flavonoid compounds, and also antioxidant activity were obtained from all celery root and leaf samples, and additionally, fiber, protein, fat, sugar, and starch contents were obtained from the celery root samples. Principal component analysis (PCA) and cluster analysis were used in order to find correlations between the different production conditions and the investigated parameters in the celery root and leaf samples.

## 2. Materials and Methods

### 2.1. Sample

The celery cultivar “Diamant” of the Bejo Zaden seed house was used in experiment (2019, from June 16 to November 27). This cultivar is characterized as having a medium length of vegetation; it forms a smooth and round thickened root of white tissue color, and it is intended for usage in its fresh state and for storage.

### 2.2. Sample Preparation

The celery was used when it reached the stage with developed thickened roots. Samples of plant material were taken for quality testing depending on fertilization with organic or mineral fertilizers. At the end of the growing season, the biological yield and the yield of the thickened roots were measured. After that, samples of thickened roots and leaves were taken for further analysis.

The root (R) and leaf (L) samples were marked with a number and letter, i.e., 1—control without fertilization; 2—mineral fertilizer, 800 kg/ha NPK fertilizer, formulation 9:12:25 (9 kg N, 12 kg P_2_O_5_, 25 kg K_2_O in 100 kg fertilizer) and 288 kg/ha AN (ammonium nitrate); 3—poultry manure, 36.5 t/ha; 4—cattle manure, 18 t/ha; 5—sheep manure, 40 t/ha; 6—supercompost, 33.9 t/ha; 7—molasses, 8.5 t/ha.

Supercompost is a sludge produced in the wastewater treatment plants of breweries. It is created by fermenting organic matter of plant origin, is gray-brown in color, has granules smaller than 25 mm, and exhibits a slightly alkaline reaction. The heavy metal content is below the maximum allowed concentration.

Molasses is the syrup that remains after the crystallization of sugar from sugar beet. The syrup is separated from the sugar crystals by centrifugation several times, and after the last stage, a thick, viscous, dark-colored product remains. It is suitable for fertilizing the soil because it has a rich content of mineral substances, and the sugar residues stimulate the activity of microorganisms.

### 2.3. Experimental Design

The experiment was set up according to a random block system in three repetitions in Futog (Serbia) on the Drljača family farm.

### 2.4. Production Conditions

During the course of the field experiment, the highest average daily temperature was recorded in August (24.32 °C), which was 3.6 °C higher than the long-term average (1981–2010) (Figure 1). The lowest temperature was measured in November (12.42 °C), which was 6.07 °C higher than the long-term average. In terms of precipitation, the highest amount was recorded in June (71.4 mm), while the lowest was in October (17.20 mm); these were 20 mm and 35.5 mm less than the long-term average, respectively.

Different amounts of fertilizers were applied to all fertilization variants (except the control), but equal amounts of 170 kg N/ha were added. This amount of N is the maximum amount that can be used in organic production after a three-year conversion period, which is prescribed by the Nitrate Directive [19].

Before fertilizing, an agrochemical analysis of the organic fertilizers and soil was performed, and the results are presented in Table 1 and Table 2.

Celery production was conducted in an open field, but seedling production was undertaken in a protected area in a tunnel-type facility without additional heating. Sowing was performed in mid-April in containers with 209 holes. The entire amount of organic fertilizer, as well as 800 kg/ha of mineral fertilizer (NPK 9:12:25), was applied one week before planting by spreading the fertilizer over the entire surface of the soil and incorporating the fertilizer with a seed drill. The remaining amount of mineral fertilizer (288 kg/ha ammonium nitrate) was applied twice during the celery vegetation. One half of the ammonium nitrate was applied after plant rooting, and the other half before the beginning of root thickening.

Sixty-five-day-old seedlings (at the stage of having 5 fully developed leaves) were planted with a spacing of 70 cm between rows and 25 cm between plants within a row, achieving a density of 5.7 plants per m^2^. The size of the basic plot was 14 m^2^, and the size of the calculation plot was 7 m^2^. All plants were used to determine the examined parameters.

### 2.5. Analysis

#### 2.5.1. Dry Weight

The content of dry weight (DW) of all celery samples was determined gravimetrically, by drying the sample at a temperature of 105 ± 0.5 °C to a constant mass [20]. All measurements were repeated in triplicate and the results are expressed as the mean moisture value in percent.

#### 2.5.2. Extract Preparation

Samples were firstly ground in a basic mill (model A11BS000, IKA, Stauffen, Germany) before extraction. The ground samples were transferred to Erlenmeyer flasks with a volume of 50 mL, covered with 25 mL of extraction solvent (methanol), and sealed with foil. After that, the Erlenmeyer flasks with the samples were placed on a shaker (UNIMAX 1010, Heidolph, Germany) and mixed for 24 h in the dark at room temperature at 100 rpm. Subsequently, the samples were quantitatively transferred into 50 mL volumetric flasks, which were then filled with extraction solvent up to a total volume of 50 mL. The contents of the flasks were then filtered through qualitative filter paper. The prepared extracts were stored in a refrigerator (4 °C) until the time of analysis and were used for the analysis of TPC content, total flavonoid content, and antioxidant activity.

#### 2.5.3. Total Phenolic Content (TPC)

The TPC of all celery samples was determined spectrophotometrically (UV/VIS-Spectrometer LLG-uniSPEC 2, LLG Labware, Meckenheim, Germany) by the Folin–Ciocalteu method [21], using gallic acid as a standard. TPC is expressed as gallic acid equivalent (mg GAE/100 g DW).

#### 2.5.4. Total Flavonoid Content (TF)

The content of TFs in all celery samples was determined spectrophotometrically, using the colorimetric method with aluminum chloride [22]. The content of TFs is expressed as catechin equivalent (mg CE/100 g DW).

#### 2.5.5. 2,2-Diphenyl-1-picryl-hydrazyl (DPPH)

The free-radical-scavenging activity of all celery samples was determined using the spectrophotometric method described by Espin et al. [23]. Prepared celery extract was mixed with methanol (96%) and 90 μM 2,2-diphenyl-1-picryl-hydrazyl (DPPH). After 60 min at room temperature, the absorbance was measured at 517 nm. Antioxidant activity is also expressed as IC_50_, which represents the concentration (extract solution) required for obtaining 50% of the radical scavenging capacity. Higher IC_50_ values correspond to lower antioxidant activity. IC_50_ values are expressed in mg/mL units.

#### 2.5.6. Fiber

The content of cellulose (crude fiber) in root celery samples was determined by the Kirschner–Ganakova method described in Vračar [24].

#### 2.5.7. Fats

The procedure for determining the fat content in root celery samples was carried out according to the Soxhlet extraction method [25], using diethylether as a solvent.

#### 2.5.8. Proteins

Protein content in the root celery samples was determined by the Kjeldahl method described in the study by Marcó et al. [26].

#### 2.5.9. Total Sugars

The content of total sugars in root celery samples was analyzed according to the Luff–Schoorl method [20]. All measurements of fat, protein, and total sugar content were repeated three times and the results are expressed as the mean value in percentages.

#### 2.5.10. Starch

The content of starch in root celery samples was determined by acid hydrolysis [24].

#### 2.5.11. Statistical Analysis

All the data were analyzed by univariate analysis of variance (ANOVA, *p* < 0.05) in order to differentiate the samples using an α = 0.05 criterion and Tukey’s Multiple Comparison Test. Principal component analysis (PCA) was applied in order to analyze and structure the obtained results. Statistica 10.0 (StatSoft Inc., Tulsa, OK, USA) was used both for the ANOVA and PCA.

## 3. Results and Discussion

### 3.1. Dry Weight (DW)

DW content is an important indicator of celery root quality. Celery root is often dried and used in various seasonings, and a higher DW enables a greater yield of dried celery from each unit of fresh root. In this study, the highest DW content was observed in sample 6R (10.52%), while the lowest was in sample 2R (9.49%), though the difference was not statistically significant. This lack of variation in DW levels may be attributed to the strong influence of celery’s genotype, which tends to dominate this trait and make it less susceptible to external factors. This finding is consistent with previous research on wheat (*Triticum aestivum* L.) and cotton (*Gossypium hirsutum* L.) [27,28]. Additionally, DW content impacts storage potential, as higher dry matter levels reduce water loss during storage.

Celery leaves, frequently used as a spice in dehydrated form, benefit from higher DW content, as this reduces the energy required to dry each kilogram of leaves. In our study, the highest DW in leaves was recorded in sample 1L (13.56%) and the lowest in sample 7L (11.33%), with this difference being statistically significant. This may be due to the N content of the fertilizer, which affects the carbon (C)-to-N ratio in cells by reducing C in favor of N, leading to a decrease in carbon-based compounds such as cellulose and an increase in water content. Similar observations were reported in a study on cucumber (*Cucumis sativus* L.) [29]. Moreover, no significant differences in DW content were observed among leaf samples 2L–7L, aligning with the findings from a study that applied sheep manure, which did not significantly affect DW content in celery leaves [30].

### 3.2. Total Phenolic Content (TPC)

Since the high importance of phenolic compounds in the human diet and their positive influence is generally well known, the contents of these compounds have been deeply investigated in various vegetable sorts and varieties, including celery root and leaves. In the investigation by Číž et al. [31], it was concluded that total polyphenol content was significantly higher in celery leaf extracts compared to all other investigated extracts, such as parsley leaves, radish, eggplant, broccoli, and many others. The same paper also showed that the polyphenol content in celery leaves (605.6 mg GAE/100 g FW) is significantly higher (*p* < 0.05) than in celery root (43.8 mg GAE/100 g FW). TPC was also investigated in the research by Priecina and Karklina [32], where the content of phenolic compounds was 330 mg GAE/100 g DW in celery leaves and 363.50 mg GAE/100 g DW in celery root. TPC in this research varied between 112.5 and 142.5 mg GAE/100 g DW and between 433.3 and 494.7 mg GAE/100 g DW in celery roots and leaves, respectively. Marinova et al. [33] quoted similar TPC in celery leaves as in this research, while Đurović et al. [34] measured significantly higher TPC in celery root. The results are presented in Table 3, and it can be noted that there is a difference in terms of TPC between celery root and leaves, i.e., the values of TPC in the leaf samples are higher, but there is no significant difference between the results in the frame of the root and leaf samples separately. Golubkina et al. [35] reported that there was a significant difference in polyphenol content between the edible parts of root celery plants, but there was no difference between varieties. Significantly different TPC values between cabbage varieties were reported Singh et al. [36]. The highest TPC in root samples was obtained in the 1R sample, i.e., in the control sample without fertilization, while the lowest content was observed in the 3R sample, where poultry manure was used for fertilization. Fertilization reduced the phenol content in cabbage heads [37], while opposite results were found in the experiments of Golubkina et al. [35] on onion, Amarowicz et al. [38] on Jerusalem artichoke, Zhao et al. [39] on Welsh onion, and Al Subeihi et al. [40] on carrot. The TPC in the samples of celery root was higher for the mineral fertilizer variant than the organic manure variants. De Oliveira Pereira et al. [41] stated, in contrast, that higher TPC was found in carrot, pepper, and lettuce in an organic system compared with a conventional system. In the case of the celery leaves, the highest TPC was obtained in the 6L sample where supercompost was implemented, while the lowest one was observed in the 2L sample, where the mineral fertilizers were used. This is in agreement with Czech et al. [42], who reported higher polyphenol content in samples of garlic, leek, yellow onion, and red onion grown under organic cultivation methods compared with conventionally grown plants.

### 3.3. Total Flavonoid (TF) Compounds

The observed range of TFs in celery root samples was between 35.6 and 53.6 mg CE/100 g DW, while this range in the case of celery leaves was significantly higher (*p* < 0.05), between 234.6 and 309.4 mg CE/100 g DW. Based on the results presented in Table 3. It can be noticed that a statistically significant difference in the mean values of TFs between the celery root and leaf samples was detected. In the research by Golubkina et al. [35], a significant difference in TFs between the edible parts of celery plants was obtained, while no significant difference was found between the tested varieties. Marinova et al. [33] found significantly lower TF content in celery leaves, while Đurović et al. [34] measured significantly higher TF content in celery roots. The highest TF value in our root samples was found in sample 7R, where molasses was used for fertilization, while the lowest TF value was observed in the same sample that had the lowest TPC, where poultry manure was used for fertilization. Similar results were reported by Al Subeihi et al. [40] in an experiment on carrots using mineral and organic fertilization. In the case of the leaves, the highest TF content was obtained in the same sample where the highest TPC was obtained, i.e., in sample 6L, where supercompost was implemented, while the lowest one was observed in sample 7L, where molasses fertilization was used. The lower content of TFs in the celery roots fertilized with poultry manure and celery leaves fertilized with molasses is a consequence of the different availability of nutrients compared to the other investigated variants. The mineralization rate of organic fertilizers depends on the type of fertilizer, the degree of organic matter decomposition, temperature, and microbiological activity, and their value as a source of nutrients for plants varies greatly [10]. Increasing the availability of N reduces the content of TFs, which was proven by Vojnović et al. [43] in experiment on onion.

### 3.4. DPPH (IC_50_)

The obtained range of antioxidant activity in the celery root samples was between 85.6 and 274.1 mg/mL, while this range in the case of celery leaves was significantly higher: between 2.15 and 3.37 mg/mL. This is in agreement with Golubkina et al. [35], who stated the significantly higher antioxidant activity of leaves compared to celery roots. The highest antioxidant activity, i.e., the lowest IC_50_ value in the celery root samples, was obtained in the 2R sample, i.e., in the sample where mineral fertilizer was used for fertilization, while the lowest antioxidant activity, i.e., the highest IC_50_ value, was observed in the 7R sample where molasses was used for fertilization. This is not in agreement with Cezch et al. [42], who achieved higher antioxidant activity in samples of garlic, leek, yellow onion, and red onion grown in an organic system rather than a conventional system where mineral fertilizers were applied. In the case of the leaves, the highest antioxidant activity, i.e., the lowest IC_50_ value, was obtained in the same sample that the highest TPC and TF values were obtained, i.e., in the 6L sample where supercompost was implemented, while the lowest antioxidant activity, i.e., the highest IC_50_ value, was observed in the 2L sample where the mineral fertilizers were used, and where the lowest TPC was also obtained. Similarly, in their research on lettuce, Kundu [44] concluded that the application of mineral fertilizer decreased the antioxidant activity, i.e., the IC_50_ value was higher. Compost applications significantly affected fruit antioxidant activity in experiments on pepper [45,46] and on strawberries [47], while opposite results were quoted for cabbage [37]. The same authors concluded that stressed treatments from a nutritional point of view showed a high content of antioxidant compounds. In Table 4, the characterization of the celery root is presented.

Table 4 shows the nutritional values of the celery root samples (1R to 7R) in terms of fiber content, total fat, protein, total sugars, and starch. The fiber content of the celery root samples varied between 16.04 g and 20.84 g. Sample 2R had the highest fiber content (20.84 g), while sample 7R was at the lower limit (16.04 g). The F-value is 3.198 and the *p*-value is 0.034, indicating that there is a statistically significant difference between the samples in terms of fiber content. Fat content ranged from 1.86 g to 3.36 g, with an average value of 2.41 g. Sample 4R had the highest fat content, while 6R had the lowest. An F-value of 28.616 and a *p*-value of 0.000 indicate a statistically highly significant difference in fat content between the samples. The protein content varied from 11.29 g to 14.77 g, with an average value of 13.02 g. Sample 2R had the highest protein content, while 7R had the lowest. The F-value (1.803) and *p*-value (0.170) show that the differences in protein content between the samples are not statistically significant. The sugar content varied from 7.48 g to 16.94 g, with sample 4R having the highest sugar content and sample 1R the lowest. The F-value (29.086) and *p*-value (0.000) indicate statistically significant differences between the samples in terms of sugar content. Starch content varied from 10.63 g to 12.57 g, with an average value of 11.75 g. No significant differences were observed between samples in terms of starch content (F = 1.666, *p* = 0.202). In short, the contents of fiber, fat, and sugar showed statistically significant differences between the samples (*p* < 0.05), while the contents of protein and starch had no significant differences. The investigation of celeriac roots in the manner investigated in this research has not been extensively studied by other authors, though it has certainly been a subject of research, particularly in recent years. For example, Daneshvar et al. [48] evaluated the impact of organic and mineral fertilizers on plant growth, minerals, and the postharvest quality of celery. This was also investigated in the paper by Godlewska et al. [49], who conducted a field-scale evaluation of the effects of botanical extracts on the yield, chemical composition, and antioxidant activity of celeriac.

### 3.5. PCA 

The correlation results of the celery root samples indicate a non-significant relationship between DPPH and both TPC and TFs in the tested root samples. The correlation coefficient between DPPH and TPC is *r* = −0.121 (*p* = 0.796), suggesting a very weak negative association that is statistically insignificant. Similarly, the correlation between DPPH and TF is positive but extremely weak (*r* = 0.044) and also statistically non-significant (*p* = 0.926).

PCA was applied to the obtained data in order to obtain a better overview of the similarities between the celery root and leaf samples based on the investigated properties of these samples (DW, TPC, TFC, IC_50_, and additionally fiber, total fat, protein, total sugars, and starch for celery root). A multivariate statistical approach, i.e., PCA, was applied in order to differentiate between the samples grown in the different production conditions and investigate the parameters of the celery root and leaf samples.

The PCA biplot of the celery root samples, illustrating the relationships between DW, TPC, TF, IC_50_, fiber, total fat, protein, total sugars, and starch, revealed that the first two principal components accounted for 68.74% of the total variance in the observed parameters (Figure 2a). Notably, the PCA showed a clear clustering pattern among the celery root samples.

Upon examining the PCA results, it is evident that TPC contributed significantly (17.4% of the total variance), along with fiber (17.0%), protein (7.0%), and starch (13.76%), all showing a positive correlation with the first principal component (PC_1_). In contrast, DW (18.59%), IC_50_ (12.4%), and total sugar (13.7%) negatively influenced PC_1_.

Furthermore, fiber (11.2% of the total variance), total fat (29.4%), protein (19.8%), and total sugar (12.3%) positively influenced the second principal component, whereas TFs (19.1%) and starch (7.4%) were associated with negative influences on PC_2_.

Sample R6 exhibited the highest DW, while sample 2R had the lowest DW. Sample 1R showed the highest TPC, while sample 3R had the lowest TPC. For TF content, sample 7R had the highest level, and sample 3R had the lowest TFs. The IC_50_ radical scavenging activity was highest in sample 7R and lowest in sample 2R. Fiber content was highest in sample 2R and lowest in sample 7R. Sample 4R had the highest total fat content, while sample 6R had the lowest. Protein content was highest in sample 2R and lowest in sample 7R. Sample 4R had the highest total sugar content, whereas sample 1R had the lowest total sugar content. Lastly, starch content was highest in sample 6R and lowest in sample 4R.

The PCA biplot analysis of the celery leaf samples revealed that the first two principal components accounted for 88.87% of the total variance in the observed parameters (Figure 2b). Notably, the PCA displayed a clear clustering pattern among the celery leaf samples.

Upon closer examination of the PCA results, it became evident that the IC_50_ value made a substantial contribution, explaining 35.99% of the total variance, and showed a positive correlation with the first principal component (PC_1_). Conversely, TPC (35.4%) and TFs (28.1%) negatively influenced PC_1_.

Additionally, DW (72.9% of the total variance) had a positive impact on the second principal component (PC_2_), while TPC (14.9%) exhibited a negative influence on PC_2_.

The correlation analysis for celery leaf samples suggests a statistically significant negative relationship between DPPH and TPC, with a correlation coefficient of *r* = −0.789 (*p* = 0.035), indicating that as TPC increases, DPPH scavenging activity tends to decrease, or vice versa. This finding suggests that TPC may play a role in modulating DPPH activity in the tested samples. In contrast, the correlation between DPPH and TFs is also negative (*r* = −0.504) but lacks statistical significance (*p* = 0.249), indicating that the relationship may be weak or inconsistent in this dataset.

The PCA of the celery leaf samples reveals notable differences in various biochemical parameters. Sample 1L has the highest DW, whereas sample L7 has the lowest DW. In terms of TPC, sample 6L exhibits the highest value, while sample 2L has the lowest TPC. Sample 6L also showed the maximum TFs, with sample L7 having the lowest TFs. For IC_50_ radical scavenging activity, sample 2L shows the highest activity and sample 6L the lowest IC_50_ value.

PCA was carried out in order to reduce the number of dimensions of the complex system with four grouping variables. It can be seen that PC_1_ and PC_2_ accounted for 97.52% (Figure 2) of the total variance of the model. PC_1_ was negatively correlated with all grouping parameters except IC_50_, while PC_2_ was negatively correlated with all investigated parameters.

The distribution of the samples was significantly influenced by the part of the celery sample, i.e., the root or leaf, as these two groups could clearly be grouped together (Figure 2a). Within the celery root samples, the following groups were identified: 2R and 5R (mineral fertilizer and sheep manure); 6R (supercompost); 1R and 3R (control and poultry manure); and 4R and 7R (cattle manure and molasses) (Figure 2b). In the case of the celery leaf samples, the following groups of samples were noticed: 1L (control); 2L, 3L, and 4L (mineral fertilizer, poultry manure, and cattle manure); 5L and 6L (sheep manure and supercompost); and 7L (molasses).

### 3.6. Cluster Analysis

Hierarchical cluster analysis was applied in this research for the same reason as PCA, i.e., in order to find correlations between the different production conditions and investigated parameters in the celery root and leaf samples.

The cluster analysis performed on the celery root samples unveiled two primary clusters, as depicted in Figure 3a. Within the first cluster were samples 1R, 2R, 3R, and 4R, where no fertilization, mineral fertilization, poultry manure, and cattle manure, respectively were used. The second cluster included samples 5R, 6R, and 7R, where sheep manure, supercompost, and molasses, respectively, were used. The linkage distance between these clusters was approximately 1600 units.

The cluster analysis performed on the celery leaf samples unveiled two primary clusters, as shown in Figure 3b. The first cluster included samples 1L and 5L, where no fertilization and sheep manure, respectively, were used. The second cluster included samples 2L, 3L, and 4L, where mineral fertilizer, poultry manure, and cattle manure, respectively, were used. Samples 6L and 7L, where supercompost and molasses, respectively, were used, were left out of the main two clusters. The linkage distance between these clusters was approximately 50 units, while the linkage distance between the main clusters and samples 6L and 7L was 125 units.

In summary, applied statistical methods (PCA and hierarchical cluster analysis) allowed us to identify and interpret the complex relationships between the various fertilization conditions implemented in this research and the observed nutritional parameters in the celery samples. On one side, PCA enables reducing the complexity of data by highlighting the main factors influencing variations in bioactive content and antioxidant activity in the frame of the samples. On the other side, hierarchical cluster analysis groups samples based on similarity, revealing patterns in how specific fertilization methods affect the composition of celery. Both statistical analyses emphasize the impact of cultivation options on optimizing celery’s nutritional and functional properties.

## 4. Conclusions

This study highlights significant differences in total TPC, TF levels, and antioxidant activity between celery leaves and roots, with leaves exhibiting higher concentrations. Cluster analysis and PCA further confirmed that these differences are associated with plant parts and agronomic practices, underscoring the importance of fertilization methods on the nutritional properties of celery. The highest fiber and protein contents were found in mineral-fertilized roots, the highest total fat and sugar contents in cattle-manure-fertilized roots, and the highest starch content in supercompost-fertilized roots. Fertilization with supercompost yielded the highest TPC and TFs in leaves, while mineral fertilizer resulted in the highest antioxidant activity in roots. Notably, the highest DW in leaves and the highest TFC and TF content in roots were also observed with supercompost. These findings suggest that supercompost, a byproduct of breweries, could potentially replace animal-based organic fertilizers, addressing the problem of reduced availability due to declining livestock numbers.

## Figures and Tables

**Figure 1 foods-13-03652-f001:**
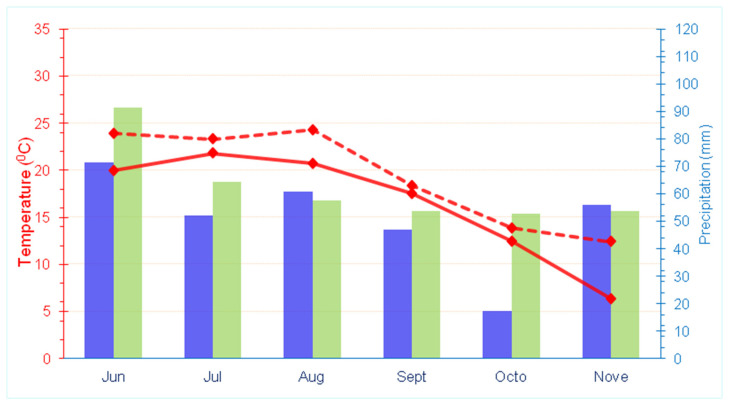
Meteorological conditions during the experiment (from June to November 2019). The blue bars represent total precipitation, while the green bars indicate the long-term average precipitation (1981–2010). The solid red line represents the average daily temperature, and the dashed red line represents the long-term average daily temperature (1981–2010).

**Figure 2 foods-13-03652-f002:**
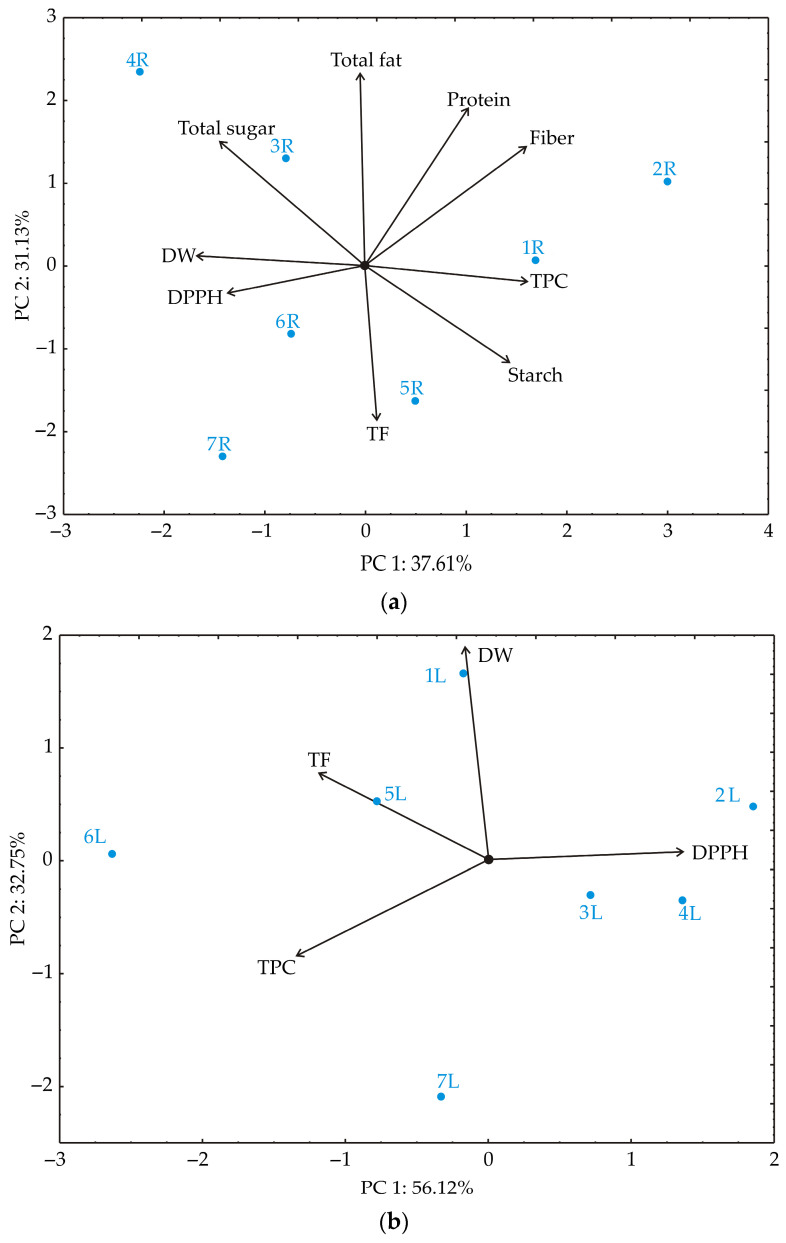
(**a**) The PCA biplot of the relationships between DW, TPC, TF, IC_50_, the contents of fiber, total fat, protein, total sugar, and starch in celery root samples. (**b**) The PCA biplot diagram of the relationships between DW, TPC, TF, and IC_50_ in celery leaf samples. The root (R) and leaf (L) samples are marked with a number and letter, i.e., 1—control without fertilization; 2—mineral fertilizer (800 kg/ha NPK fertilizer formulation 9:12:25 and 288 kg/ha AN); 3—poultry manure, 36.5 t/ha; 4—cattle manure, 18 t/ha; 5—sheep manure, 40 t/ha; 6—supercompost, 33.9 t/ha; 7—molasses, 8.5 t/ha.

**Figure 3 foods-13-03652-f003:**
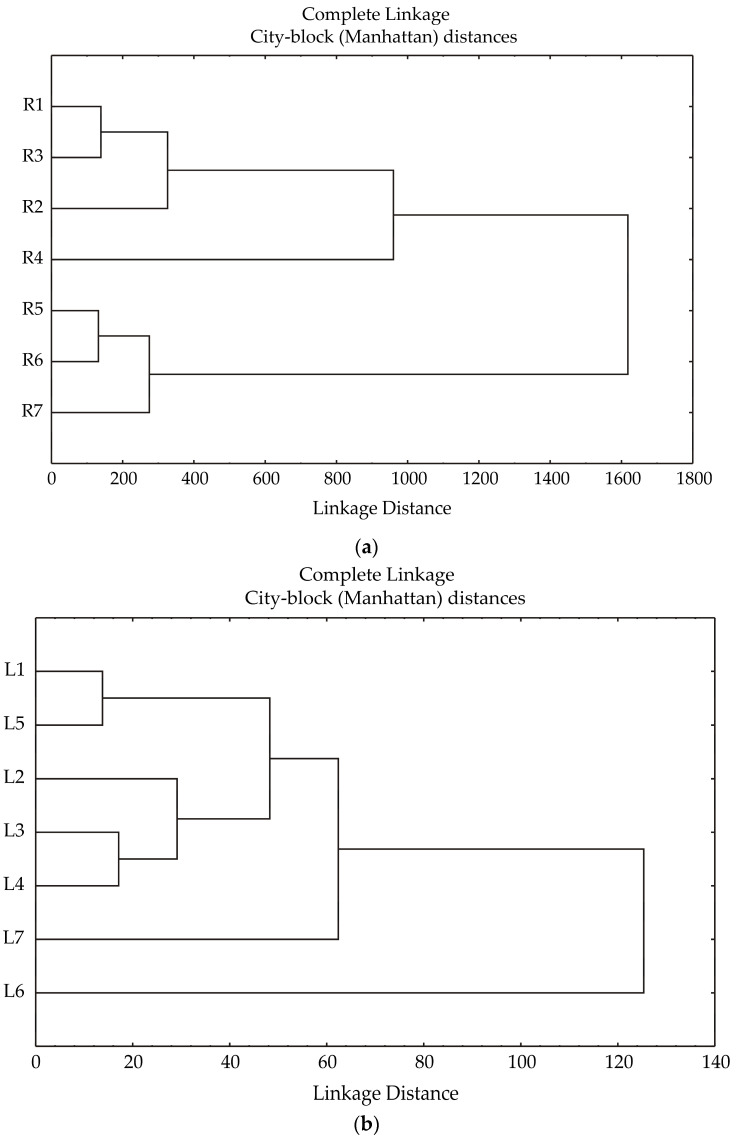
(**a**) The cluster analysis of the celery root samples, according to DW, TPC, TF, IC_50_, fiber, total fat, protein, total sugar, and starch. The root (R) samples are marked with a number and letter. (**b**) The cluster analysis of the celery leaf samples, according to DW, TPC, TF, and IC_50_. The leaf (L) samples are marked with a number and letter, i.e., 1—control without fertilization; 2—mineral fertilizer (800 kg/ha NPK fertilizer formulation 9:12:25 and 288 kg/ha AN); 3—poultry manure, 36.5 t/ha; 4—cattle manure, 18 t/ha; 5—sheep manure, 40 t/ha; 6—supercompost, 33.9 t/ha; 7—molasses, 8.5 t/ha.

**Table 1 foods-13-03652-t001:** Results of agrochemical analysis of organic fertilizers (%).

Organic Fertilizer	N	P_2_O_5_	K_2_O	Moisture	Organic Matter	Organic Carbon
Sheep manure	1.76	1.01	2.64	75.94	67.24	24.09
Supercompost	2.21	3.05	0.13	77.29	41.71	15.19
Cattle manure	2.25	3.83	2.22	58.08	48.95	16.25
Poultry manure	1.00	1.56	1.69	53.45	28.02	14.90
Molasses	2.00	/	5.0	/	/	16.50

**Table 2 foods-13-03652-t002:** Results of soil chemical analysis (soil depth 0–30 cm).

pH		CaCO_3_ (%)	Humus (%)	Total N (%)	Al-P_2_O_5_ (mg/100 g)	AL-K_2_O (mg/100 g)
KCl	H_2_O					
7.39	8.38	6.36	2.25	0.167	13.98	19.83

**Table 3 foods-13-03652-t003:** Experimentally obtained results on celery root (R) and leaf (L).

Sample	DW	TPC	TF	DPPH (IC_50_)
1R	9.61 ± 0.49 ^a^	142.5 ± 15.7 ^a^	37.0 ± 4.2 ^ab^	153.9 ± 34.6 ^b^
2R	9.49 ± 0.45 ^a^	133.9 ± 11.4 ^a^	48.5 ± 0.5 ^abc^	85.6 ± 8.8 ^a^
3R	10.02 ± 0.64 ^a^	112.5 ± 13.3 ^a^	35.6 ± 3.5 ^a^	148.2 ± 2.5 ^b^
4R	10.23 ± 0.57 ^a^	115.3 ± 6.0 ^a^	39.6 ± 5.7 ^ab^	211.8 ± 1.1 ^c^
5R	9.70 ± 0.47 ^a^	118.7 ± 14.3 ^a^	49.8 ± 5.3 ^bc^	99.1 ± 0.1 ^a^
6R	10.52 ± 0.01 ^a^	117.3 ± 3.8 ^a^	47.8 ± 4.5 ^abc^	102.7 ± 12.3 ^a^
7R	9.96 ± 0.29 ^a^	123.4 ± 3.4 ^a^	53.6 ± 6.8^c^	274.1 ± 2.1 ^d^
**Average**	**9.93 ± 0.52**	**123.4 ± 3.8**	**44.6 ± 7.8**	**153.7 ± 66.1**
F	1.870	3.066	6.647	68.513
*p*	0.157	0.039	0.002	0.000
1L	13.56 ± 0.37 ^b^	449.9 ± 56.6 ^a^	264.1 ± 37.1 ^ab^	2.44 ± 0.30 ^ab^
2L	12.51 ± 0.67 ^ab^	433.3 ± 32.8 ^a^	246.7 ± 3.1 ^a^	3.37 ± 0.03 ^c^
3L	12.31 ± 1.06 ^ab^	456.4 ± 38.2 ^a^	241.5 ± 0.3 ^a^	2.84 ± 0.36 ^bc^
4L	12.04 ± 0.70 ^ab^	447.8 ± 10.0 ^a^	249.2 ± 13.1 ^a^	3.31 ± 0.28 ^c^
5L	12.85 ± 0.52 ^ab^	462.7 ± 15.0 ^a^	264.0 ± 10.5 ^ab^	2.17 ± 0.09 ^a^
6L	12.44 ± 0.37 ^ab^	494.7 ± 5.0 ^a^	309.4 ± 4.0 ^b^	2.15 ± 0.04 ^a^
7L	11.33 ± 0.12 ^a^	480.5 ± 30.5 ^a^	234.6 ± 19.7 ^a^	2.32 ± 0.04 ^ab^
**Average**	**12.43 ± 0.83**	**460.7 ± 33.1**	**258.5 ± 27.7**	**2.66 ± 0.52**
F	3.795	1.297	6.234	18.415
*p*	0.02	0.320	0.002	0.002

Values marked with the same letter in a column are not significantly different at 5% (Tukey’s HSD test). The root (R) and leaf (L) samples are marked with a number and letter, i.e., 1—control without fertilization; 2—mineral fertilizer (800 kg/ha NPK fertilizer formulation 9:12:25 and 288 kg/ha AN); 3—poultry manure, 36.5 t/ha; 4—cattle manure, 18 t/ha; 5—sheep manure, 40 t/ha; 6—supercompost, 33.9 t/ha; 7—molasses, 8.5 t/ha. DW—dry weight (%); TPC—total phenolic content (mg GAE/100 g DW); TF—total flavonoid content (mg CE/100 g DW); IC_50_—antioxidant activity (mg/mL).

**Table 4 foods-13-03652-t004:** Characterization of the celery root.

Sample	Fiber (g/100 g DW)	Total Fat (g/100 g DW)	Protein (g/100 g DW)	Total Sugar (g/100 g DW)	Starch (g/100 g DW)
1R	17.96 ± 1.30 ^ab^	2.56 ± 0.92 ^b^	12.94 ± 0.04 ^a^	7.48 ± 0.08 ^a^	12.20 ± 1.73 ^a^
2R	20.84 ± 3.16 ^b^	2.69 ± 0.76 ^b^	14.77 ± 1.69 ^a^	8.73 ± 0.12 ^ab^	12.32 ± 1.77 ^a^
3R	18.10 ± 0.62 ^ab^	2.47 ± 1.50 ^b^	13.40 ± 2.01 ^a^	11.32 ± 0.38 ^bc^	10.86 ± 1.16 ^a^
4R	17.30 ± 0.37 ^ab^	3.36 ± 2.39 ^c^	13.32 ± 1.84 ^a^	16.94 ± 2.41 ^d^	10.63 ± 0.29 ^a^
5R	16.91 ± 0.61 ^ab^	1.98 ± 0.29 ^a^	12.08 ± 0.77 ^a^	9.15 ± 0.16 ^ab^	12.48 ± 0.31 ^a^
6R	16.41 ± 1.97 ^a^	1.86 ± 0.99 ^a^	13.33 ± 0.96 ^a^	12.60 ± 1.34 ^c^	12.57 ± 0.95 ^a^
7R	16.04 ± 0.57 ^a^	1.90 ± 0.44 ^a^	11.29 ± 1.53 ^a^	8.41 ± 0.17 ^ab^	11.20 ± 0.12 ^a^
**Average**	**17.64 ± 1.99**	**2.41 ± 0.53**	**13.02 ± 1.58**	**10.66 ± 3.24**	**11.75 ± 1.21**
F	3.198	28.616	1.803	29.086	1.666
*p*	0.034	0.000	0.170	0.000	0.202

Values marked with the same letter in a column are not significantly different at 5% (Tukey’s HSD test). The root (R) and leaf (L) samples are marked with a number and letter, i.e., 1—control without fertilization; 2—mineral fertilizer (800 kg/ha NPK fertilizer formulation 9:12:25 and 288 kg/ha AN); 3—poultry manure, 36.5 t/ha; 4—cattle manure, 18 t/ha; 5—sheep manure, 40 t/ha; 6—supercompost, 33.9 t/ha; 7—molasses, 8.5 t/ha.

## Data Availability

The original contributions presented in the study are included in the article, further inquiries can be directed to the corresponding authors.

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
