# Peer review of "Cluster and Principal Component Analyses of the Bioactive Compounds and Antioxidant Activity of Celery (*Apium graveolens* L.) Under Different Fertilization Schemes"

_foods, 2024, doi:10.3390/foods13223652_

Round 1

Reviewer 1 Report

Comments and Suggestions for Authors

The document is interesting, however, some comments should be considered for improve the document.

Line 109 to 113. Authors mentioned the fertilizer formulation (Table 1), but the Table 1 indicates other information.

Line 134 to 135. The table 2 presents information that is difficult to understand. Ej. Why the pH indicates KCl and H2O?

Line 155 to 164. The information is not correct, due to the authors indicate the extraction of the samples and do not the TPC.

Line 165. What is the name, model, etc of the spectrophotometer equipment?

Line 239. The title of the table 3 is not correct, due to the results are moisture, and bioactive compounds.

What is the objective of putting the average, el value of F and p in table 3, and 4?

The discussion of moisture in the samples is not presented.

Line 271-272. The value 274.2 and 2.16 are not correct, the value should be 274.1 and 2.15. Could the authors review the information “The obtained range of antioxidant activity in celery root samples was between 85.6 271 and 274.2, while this range in case of celery leaves was significantly higher (p < 0.05) 272 between 2.16 and 3.37.”

In Table 4 discussion is not presented.

Why only characterization of the celery root (table 4) is presented, and the leaves?

Reviewer 2 Report

Comments and Suggestions for Authors

Bioactive compounds and antioxidant activity of celery (Apium graveolens L.) submitted to different production conditions: PCA and Cluster Analysis

I write you in regard to manuscript # foods-3300600 entitled "Bioactive compounds and antioxidant activity of celery (Apium graveolens L.) submitted to different production conditions: PCA and Cluster Analysis" which you submitted to the foods.

Authors need to follow the following instructions to improve this manuscript

1)      The authors should precise the title.

2)      Page 4, line 144-146. Please add age of seedling /day after sowing

3)      Abstract should rewrite by adding significant data.

4)      DW result and discussion is missing. The authors should add.

5)      2.4.4. Antioxidant activity change to 2,2-diphenyl-1-picryl-hydrazyl (DPPH)

6)      Table 3: IC50 change to DPPH (IC50)

7)      3.3. Antioxidant activity (IC50) change to DPPH (IC50)

8)      Table 4: Result and discussion is missing. The authors should add.

9)      Conclusion should rewrite according to the best findings and recommendation within one paragraph.

10)  References: should follow the journal guideline. I have seen somewhere full journal name and somewhere abbreviated form.

11)  English grammar should check carefully.

12)  Please check carefully before resubmission.

I recommend improving the manuscript and resubmitting.

Comments on the Quality of English Language

 The English could be improved to more clearly express the research.

Reviewer 3 Report

Comments and Suggestions for Authors

There are many reports in the literature on the effect of fertilizer type on the content of bioactive compounds in celery. What makes this work stand out from others? For example:

Golubkina, N. A., Kharchenko, V. A., Moldovan, A. I., Koshevarov, A. A., Zamana, S., Nadezhkin, S., ... & Caruso, G. (2020). Yield, growth, quality, biochemical characteristics and elemental composition of plant parts of celery leafy, stalk and root types grown in the northern hemisphere. Plants, 9(4), 484.

Daneshvar, H., Babalar, M., Díaz-Pérez, J. C., Nambeesan, S., Delshad, M., & Tabrizi, L. (2023). Evaluation of organic and mineral fertilizers on plant growth, minerals, and postharvest quality of celery (Apium graveolens L.). Journal of Plant Nutrition, 46(8), 1712-1729.

Godlewska, K., Pacyga, P., Michalak, I., Biesiada, A., Szumny, A., Pachura, N., & Piszcz, U. (2020). Field-scale evaluation of botanical extracts effect on the yield, chemical composition and antioxidant activity of celeriac (Apium graveolens L. var. rapaceum). Molecules, 25(18), 4212.

The literature review is based on fairly old literature. There are many reports from the last 4 years that can be used.

The paper should describe in detail the conditions in which the celery was grown and clearly state that it was grown in the same conditions, and the only variable factor was the type of fertilizer used. This is not described in the paper, and otherwise the assumptions of the paper would not make sense.

Pure methanol was used to extract polyphenols. As is known from various studies, this is a very ineffective solvent for this type of compounds. The best solvents are those mixed with water - methanol 80% or acetone 80%. The extraction efficiency using an aqueous solution of methanol is much higher. Please explain why one of the least effective variants was chosen? The aim of the work was to prove the antiradical potential and its variability depending on the type of fertilization used, hence the most optimal extraction conditions had to be used.

The determination of total polyphenols and total flavonoids is very non-specific. In this work, the HPLC or LCMS method should be used to assess the change in these compounds.

Reviewer 4 Report

Comments and Suggestions for Authors

This research investigates the impact of various fertilization methods on the bioactive compound content and antioxidant activity of celery (Apium graveolens L.) root and leaf. Principal Component Analysis (PCA) and Cluster Analysis were used to correlate production conditions with phenolic and flavonoid content, and  antioxidant activity, between celery root and leaves and cultivation conditions. It was found that the byproduct of breweries, could potentially replace animal-based organic fertilizers. The manuscript is well written. The rationale of this study is relevant to food chain and food processing.

1. To examine the quality of the plant samples, only total phenolic and total flavonoid contents are not sufficient. The analysis of known reference compounds found in the samples must be elucidated by both quantitative and quantitative analyses. 

2. Fiber, protein, sugar and starch levels tested in this study should be compared with previous reports that cultivated the plant using conventional or other conditions.

3. It would be nice to separate  the paragraph of Sample preparation from Section 2.1 (as Section 2.2.)

4. Please briefly describe the Method of Section 2.4.2 and Section 2.4.9.

5. It would be benefit for readers if the author provide a brief rationale of using PCA and cluster analysis in this study.

6. To analyze anti oxidant activity, at least 2  assays with different principle must be determined. 

7. I do agree that the supercompose  could be used to replace animal-based organic fertilizers. Could you please discuss more about the processing  (is it simple?) and also the cost of the production of the supercompost compare to animal-based organic fertilizers.

Round 2

Reviewer 1 Report

Comments and Suggestions for Authors

Thanks you, the comments were considered.

Author Response

The author thanks for the reviewer's comment.

Reviewer 2 Report

Comments and Suggestions for Authors

The authors should improve this manuscript:

1) Check ml or mL. I think use mL entire manuscript.

2) In line 99 add experiment conducting year and months.

3) In 2.4 Production conditions, why used 1981-2010 meteorological data and graph? Add research time data/graph with the line indicator. 

Author Response

The authors would like to thank the Reviewer for all the constructive and helpful comments and suggestions. Based on the comments, revisions have been made to the corrected manuscript, and all responses to the questions are provided in this file. Thanks to the reviewer's comments, the paper has been significantly improved.

R2.1. Check ml or mL. I think use mL entire manuscript.

Response: Thank you for the comment. This has been corrected in the entire manuscript and mL are now stated in the revised version of the paper.

R2.2. In line 99 add experiment conducting year and months.

Response: Thank you for the comment. This has been added in the revised paper.

R2.3. In 2.4 Production conditions, why used 1981-2010 meteorological data and graph? Add research time data/graph with the line indicator.

Response: The thirty-year average data were included to evaluate any significant deviations during the study year, which is standard practice in agronomic experiments. Precisely, long-term meteorological averages serve as a reference to determine if conditions in the study year deviate from the norm. This comparison allows researchers to identify potential climatic variations or extreme weather events that could influence the experiment's outcomes. By understanding whether results are typical or impacted by specific year conditions, the research conclusions gain both quality and broader relevance. Corrections regarding this Reviewer’s comment have been added in the description of the Figure 2.

Reviewer 3 Report

Comments and Suggestions for Authors                   I accept the authors' translation, although I believe that if one wants to compare results with other authors and wants to know what potential the plants studied really have, one should use methods that are best suited for this. In connection with the authors' translation, the discussion of results should also be rewritten - determinations in other conditions make it impossible to directly compare results.  

Author Response

The authors would like to thank the Reviewer for the constructive comment. Response to the question is provided in this file.

R3.1. I accept the authors' translation, although I believe that if one wants to compare results with other authors and wants to know what potential the plants studied really have, one should use methods that are best suited for this. In connection with the authors' translation, the discussion of results should also be rewritten - determinations in other conditions make it impossible to directly compare results.

Response: Thank you for this valuable comment. However, when experiments are conducted in open-field conditions across different years and locations, varying results are expected. What is crucial are the consistent trends and patterns observed, which can be compared with similar studies. Thus, we are not comparing specific numerical values but rather the trends that emerge from these treatments and agroecological conditions. These trends provide a meaningful basis for comparison with findings from other authors.

Reviewer 4 Report

Comments and Suggestions for Authors

Most of comments are well response by the authors. I am really appraise the willing of the author to improve the manuscript.

However, two critical comments have been being concerned.

1. Phytochemical analysis of known reference compound(s) predominantly found in the plant was not elucidated. Moreover, the correlation of total phenolic and flavonoid contents with anti-oxidant activity must be determined, if the author would like to claim the benefit of the phytochemicals in the plant sample.

2. Only DPPH assay may not sufficient to determine anti-oxidant activity. Why the author mainly focus on only the scavenging of DPPH radical which dose not mimic to the physiological condition? Please explain.

Author Response

The authors would like to thank the Reviewer for all the constructive and helpful comments and suggestions. Based on the comments, revisions have been made to the corrected manuscript, and all responses to the questions are provided in this file. Thanks to the reviewer's comments, the paper has been significantly improved.

Authors appreciate Reviewer’s opinion and are thankful for the comments and suggestions. Firstly, this is an answer which regards to the both Reviewer’s questions.

The main reasons were outlined in the response to the reviewer in the previous review round. However, we would like to provide additional clarification regarding the selection of quality parameters for the celery samples. Namely, the experiment was designed with one of the primary objectives of comparing celery samples grown using different methods. Thus, the focus was on the comparison of these samples, rather than conducting an exhaustive analysis and chemical profiling of celery samples. Based on this approach, the authors selected the key analyses that they believe are both adequate and valid, especially considering the positive prior experiences with similar experiments published in top international journals. Therefore, in addition to the explanations provided in the previous review round, we would like to emphasize that the primary focus of our response is the comparability of the samples.

R4.1. Most of comments are well response by the authors. I am really appraise the willing of the author to improve the manuscript.

However, two critical comments have been being concerned.

Phytochemical analysis of known reference compound(s) predominantly found in the plant was not elucidated. Moreover, the correlation of total phenolic and flavonoid contents with anti-oxidant activity must be determined, if the author would like to claim the benefit of the phytochemicals in the plant sample.

Response:

Regarding the correlation of total phenolic and flavonoid contents, the correlation between total phenolic and flavonoid content and antioxidant activity have been illustrated in Figures 2a and 2b. The correlation analysis for celery leaves samples, suggest a statistically significant negative relationship between DPPH and TPC, with a correlation coefficient of r = - 0.789 (p = 0.035), indicating that as TPC increases, DPPH scavenging activity tends to decrease, or vice versa. This finding suggests that TPC may play a role in modulating DPPH activity in the tested samples. In contrast, the correlation between DPPH and TF is also negative (r = - 0.504) but lacks statistical significance (p = 0.249), indicating that the relationship may be weak or inconsistent in this dataset.

The correlation results for celery root samples indicate a non-significant relationship between DPPH and both TPC and TF in the tested root samples. The correlation coefficient between DPPH and TPC is r = -0.121 (p = 0.796), suggesting a very weak negative association that is statistically insignificant. Similarly, the correlation between DPPH and TF is positive but extremely weak (r = 0.044) and also statistically non-significant (p = 0.926).

Additional explanations about these correlations have been added in the revised paper according to the Reviewer’s suggestion.

R4.2. Only DPPH assay may not sufficient to determine anti-oxidant activity. Why the author mainly focus on only the scavenging of DPPH radical which dose not mimic to the physiological condition? Please explain.

Response: Authors want to thank the Reviewer for this valuable observation. We selected the DPPH assay as it is widely recognized in scientific literature as a standard, allowing for reliable cross-comparisons with a broad range of other studies. Also, the emphasis on the DPPH assay was chosen due to its broad application as a rapid, cost-effective method for assessing radical scavenging activity, providing a reliable benchmark for comparative purposes across studies. While DPPH does not completely mimic physiological conditions, it provides a robust measure of radical scavenging capacity and is especially valued in phytochemical research for its strong correlation with phenolic and flavonoid content, which were central compounds in this study. Including DPPH allows us to reliably assess the antioxidant potential in a way that directly supports our research focus and enhances the study’s relevance within the wider scientific context.